# Fire Safety in High-Rise Buildings: Is the Stay-Put Tactic a Misjudgement or Magnificent Strategy?

**Andrew Oyen Arewa** [1,*] **, Abdullahi Ahmed** [2] **, David J. Edwards** [3] **and Chizaram Nwankwo** [4]

1   Engineering and Construction Department, School of Architecture Computing and Engineering, Docklands Campus, University of East London, London E16 2RD, UK
2   Building and Civil Engineering Department, School of Energy Construction and Environment, Faculty of Computing and Engineering, Coventry Campus, Coventry University, Coventry CV1 5FB, UK; ab0393@coventry.ac.uk
3   School of Engineering and Built Environment, Faculty of Computing Engineering and the Built Environment, Birmingham City University, Birmingham B4 7XG, UK; david.edwards@bcu.ac.uk
4   External Quality Assessment Services, National Infection Service, Public Health England, London NW9 5EQ, UK; Nwankwo.chizaram@yahoo.com
*   Correspondence: a.o.arewa@uel.ac.uk

**Abstract:** Historically, fire incidents in high-rise buildings reveal that Fire and Rescue Services frequently rely on the stay-put tactic (i.e., occupants of high-rise buildings should remain in their apartments) during an inferno. Recent fire occurrences in high-rise buildings reveal that there are two opposing viewpoints on the stay-put tactic. First, the understanding that the stay-put tactic is a beneficial practice used to protect, control, and facilitate smooth evacuation of occupants during fire incidents. Second, the argument that the stay-put tactic is a misjudgement and futile strategy that leads to fatalities, particularly in high-rise buildings. The aim of this study was to provide awareness and understanding of fire and rescue services use of the stay-put tactic in high-rise buildings. We attempted to answer the questions: is the stay-put tactic a misjudgement or magnificent strategy? The study adopted phenomenological research strategies with various focus groups consisting of seasoned firefighters and survivors with first-hand accounts of stay-put instructions in high-rise buildings. The study also scrutinised three case studies of fire incidents in high-rise buildings in two countries. The study revealed that the stay-put tactic is obsolete; with the potential to cause catastrophic misjudgement, mostly during conflagrations in high-rise buildings. There is a need to advance research on the use of artificial intelligence communication systems and infrared image detectors camera to enhance quick and smooth fire evacuation in high-rise buildings.

**Keywords:** fire and rescue services; high-rise buildings; stay-put tactic; fire safety

## 1. Introduction

Globally, studies about fire incidents in high-rise buildings reveal catalogue of weaknesses and inconsistencies; particularly, the use of the stay-put tactic as a fire safety action plan during emergency decision-making [1]. For example, the Grenfell Tower fire in London 2017, Sao Paulo Brazil Tower fire in 2018 and many other disastrous fire incidents in buildings have one thing in common; "building occupants were given directives to stay-put in their apartments during the inferno" [2]. Recent fire incidents particularly in high-rise buildings suggest that the decision to stay-put led to preventable fatalities [2]. For instance, the UK government Grenfell Tower Inquiry [2] stated that " . . . more lives could have been saved had the stay-put decision by London Fire Brigade was abandoned as soon as it became apparent that the fire was spreading rapidly and uncontrollably . . . ".

Similarly, the National Institute of Standard and Technology [3] report on the Trade Centre Twin Towers incidents in New York City, 11 September 2001claim that "though active fire safety systems (sprinklers, smoke purge, and fire alarms) were designed to

exceed current practice . . . emergency services occupant communications i.e., instruction to remain in the buildings played significant role in the number of casualties in the 11 September 2001disaster". Though the twin towers incident was a terrorist attack, Xin and Huang [4] argued that building occupants who obeyed the stay-put directive suffered high casualties and those who evacuated the building immediately survived the disastrous incident. Other examples of ineffective fire safety decision making relating to the stay-put tactic include the 2005 Spain Torre Windsor tower fire and 2009 Beijing Television Cultural Centre inferno [1].

Conversely, Xin and Huang [4] argue that the use of the stay-put tactic suggests that the entire management of fire safety risk in high-rise buildings is not thoroughly understood. Zhang [5] opine that from "operational perspective stay-put tactic is a magnificent strategy, that protect, control and facilitate orderly evacuation of building occupants" during inferno. The United States of America's Compartment Fire Behavior Training CFBT [6] asserts that the stay-put tactic can be a useful instrument that is contingent on the nature of building and fire. However, inappropriate fire safety and evacuation decisions are often exacerbated by ignorance about the rate of fire spread and the fuel loading of different construction materials. Boyce [7] avows that sometimes, fleeing fire situation in buildings is the best response, no matter the level of emergency communication and tactics. In contrast, Fishlock [8] stress that stay-put is a mere guidance (and not a tactic) that can be best described as "a futile strategy that often leads to fatalities" particularly in high-rise buildings.

Ronchi and Nilsson [9] assert that "there is no universally accepted emergency services and occupant's communication rules for fire evacuation in buildings". In practice, Fire and Rescue Services make dynamic decisions under conditions of stress, panic and uncertainty, therefore 'stay put' as a fire safety plan of action can sometimes lead to confusion or a terrible outcome. Yet, there is little research regarding awareness and understanding of issues surrounding "why" and "how" effective the stay-put tactic is. Therefore, the aim of the study was to provide awareness and understanding of the stay-put tactic. For better understanding of the research topic, there is a need for thorough literature review about fire safety, regulations, emergency decision making and high-rise buildings.

### 1.1. Literature Review

A review of contemporary literature regarding fire in high-rise buildings reveals flaws in emergency decision making, ambiguous fire safety regulations and choice of building materials. For example, many high-rise building fire reports cite poor evacuation plans, obscure emergency response/decision making, use of inappropriate fire retardant building materials, lack of clear fire safety regulations or combination of these factors as reasons for high casualties and wanton destruction of properties, such as those relating to: the Winecoff Hotel fire in Atlanta, USA, which led to 119 casualties in 1946; the Daeyongak Hotel fire incident in Seoul, South Korea which claimed 163 lives in 1971; the MGM Grand Hotel fire in Las Vegas, USA which killed 84 victims in 1980; the Plasco Building fire in Iran which caused 21 fatalities in 2017; the Dhaka garment factory fire which led to 112 deaths in 2013; and the recent Grenfell Tower Fire, London, UK in 2017 which caused 72 fatalities. The National Fire Protection Association NFPA [10] claims that fire incidents are common to high-rise structures; but incorrect emergency services response and poor communication with building occupants often exacerbate the rate of fatalities. Therefore, it is important to provide thorough awareness and understanding of emergency services' communication to building occupants; particularly the use of the stay-put tactic.

### 1.2. Review of Key Phrases

**Stay-Put:** The phrase stay-put was first used in a Canadian judicial review to suggest that when a fire occurs within one dwelling (or less likely, in the common part); it is usually safe for residents to remain in their flats, i.e., the section of the building not affected

by fire [8]. The guidance specifically refers to false alarm cases, and blocks of flats with functional fire-proof doors and active fire safety devices.

**Fire Safety:** refers to precautions intended to prevent or reduce the likelihood of fire that may lead to adverse safety incidents or destruction of properties (Occupational Safety and Health Administration OSHA [11]. The UK Health and Safety Executive HSE [12] claim that the key concept behind fire safety is to prevent ignition of an uncontrolled fire; it also refers to other measures used to limit the development and effects of a fire after it starts. Fire safety measures are extensive because they include planned procedures undertaken during construction and operation stages of buildings, and training for the occupants of buildings. Xin and Huang [4] argue that fire safety is a significant component of buildings, yet catastrophes caused by fire around the globe suggest that the entire management of fire risk is not thoroughly understood.

**Emergency Decision Making:** refers to a snap decision taken when an emergency (such as fire outbreak, car accident, etc) occurs [13]. Emergency decision making is often characterised by time pressure and lack of information, and in some cases such decision may result in potentially serious consequences [14]. Case studies of fire in buildings show that emergency decisions usually entail fire and rescue services' chain of command having to instruct its operatives on specific communication to building occupants to evacuate using safe exit routes, the stay-put tactic and other initiatives.

**Evacuation and Escape Strategies in Buildings:** The National Fire Chiefs Council (NFCC) [15] Fire Central Programme Office refer to a fire evacuation and escape strategy as an arrangement that involves action of fire operatives when fire occurs in a building, alarm systems, escape routes, signage, emergency doors, fire-fighting equipment, fire-alarm locations, Personal Emergency Evacuation Plans (PEEPs), assembly points, etc. Evacuation strategies in buildings vary depending on the type of building. Some buildings have simultaneous evacuation policies when hearing an alarm, some maintain a Stay-Put or defence in place policy and others adopt a vertical phased approach. The Local Government Association LGA [16] fire evacuation guidance brochures title "The Fire Safety in Purpose-Built Block of Flats" claim that Stay-Put policy may be considered appropriate in a block of flats; that is, high-rise buildings based on the levels of fire resistance for compartment walls and floors. Similarly, NFCC [15] guidance state that evacuation, escape or stay-put defence in place policy to be adopt when fire occurs in a building are dependent on the following:

- Clear passageway to all evacuation routes;
- Exposure to possible hazards;
- Risks to people exiting firefighting access routes;
- People in the building (mental and physical alertness of people);
- Level of fire safety signage that is clearly marked exit routes;
- Adequate and sufficient exit routes available for all people;
- Effectiveness of emergency doors for easy evacuation;
- Effectiveness of emergency lighting provided where needed;
- Level of training when using evacuation routes;
- Clear designation of safety assembly point and level of communication.

Arguably, both LGA [16] and NFCC [15] fire evacuation strategies fail to consider, among other things, the rapid rate of fire spread, (please refer to Table S1 in Supplementary Materials) especially in modern construction materials, stairwell and lobby protection systems, smoke inhalation, confusion that often ensues in the mind of building occupants and psychology of human misjudgement when fire incidents escalate in high-rise buildings.

*1.3. Fire Safety Codes and Regulations*

Surprisingly, journal papers concerning fire evacuation plans, particularly the stay-put tactic, are scarce. However, the BS 999 code of practice for fire safety in design, management/use of buildings and the OSHA Emergency Preparedness and Response management documents are the main authoritative fire evacuation documents. However, the documents only provide partial guidance for safety training, organising efficient evacuation plans,

and allocating leadership responsibilities. Arguably, the codes do not provide specific and sufficient awareness about stay-put tactics. The only official government document that explains the stay-put tactic in apartments is produced by the UK Local Government Association [16] titled "Fire Safety in Purpose-built Blocks of Flats". The document clearly states that " ... it is normally safe for other residents to remain (stay-put) within their own flat ... " when fire breaks out in another part of the same building. Apart from the LGA [16] document, fire safety codes and regulations exist in nearly every country; some are comprehensive while others are extremely basic, if not primitive. Antell [17] argued that the development of specific fire codes and regulations for high-rise buildings began after the second world war. Literature review of fire safety codes and regulations from the late 19th century until the present day reveal six fundamental requirements as illustrated in Figure 1 below.

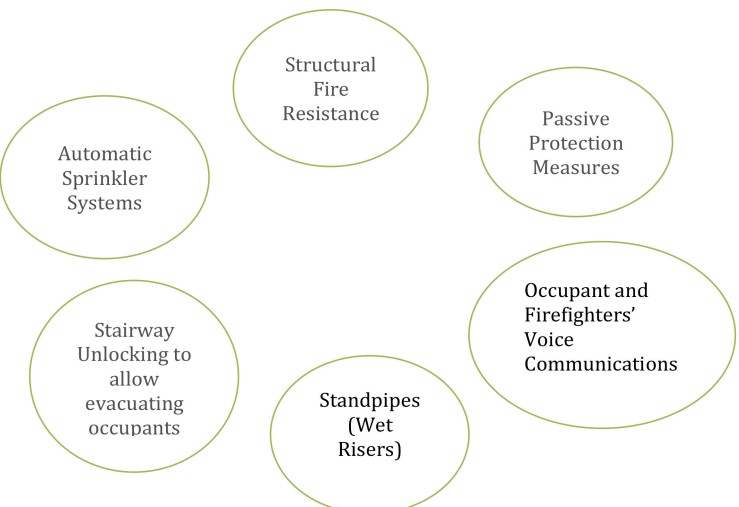

**Figure 1.** Six fundamental pillars of fire safety in Buildings.

The six significant fire safety requirements for high-rise buildings highlighted in Figure 1 have led to variations in different geographical regions. For example, different countries have specific structural fire resistance, stairway, standpoint and automatic sprinkler systems requirements. However, "occupant and firefighter voice communication" requirements, such as a stay-put plan of action, are often used blankly in most countries with little or no research on their effectiveness.

The possible reason for the blanket use of stay-put plan of action is that there is no unified fire safety legislation regarding "occupant and firefighter voice communication" requirements. For instance, central governments, states, local government authorities, and municipalities in a country usually create and rely on fire safety legislation tailored to their needs. OSHA [18] state that national and international fire safety regulations often lead to conflict due to the inadequacy of some fire safety codes, guidance or standards. Moreover, many fire codes, guidance or standards emanate from judicial precedents and in some cases fire safety professionals usually raises concern about their usefulness. For example, the Grenfell Tower Inquiry GTI [19] report claimed that " ... 72 deaths could have been saved had the stay-put directive for occupants to remain indoor during the inferno) been abandoned sooner". Fire professionals believe that the 'stay put' plan of action has been widely used up to the present day by many fire and rescue services. Yet, such plan is deemed obsolete in contemporary fire safety management; particularly in high-rise buildings. Fishlock [8] stress that stay-put fire guidance dates back to the 1960s and its origin can be traced to Canada where fire evacuation of an office block due to a false fire alarm caused panic and the ensuing rush in the building staircase caused two deaths. Subsequently, a Canadian judicial review questioned the actual need to evacuate a building with built-in fire compartmentation.

Moreover, some fire safety experts believe that fire safety laws in many countries need to be reviewed to make them relevant for modern reality. A review of UK fire laws shows that there are over 56 different fire safety codes, guidance, standards and legislations. For example, the Health and Safety Executive (HSE) fire safety guidance, the British code of practice for fire safety (BS 9999), the Regulatory Reform (Fire Safety) Order 2005, and different regional Fire and Rescue Services' (FRSs) guidance. In the USA the National Fire Protection Association (NFPA) and the America National Standards Institute (ANSI) coordinate, diffuse and create fire safety codes, standards and guidance. The NFPA alone have over 275 fire codes and standards. Moreover, there are international fire standards, for example, the European Union (EU) fire safety guide and ISO 13.220.01 Protection against Fire in General. Fishlock [8] argued that there is reason to believe that array of fire safety legislations has tendency to cause confusion particularly in emergency decision-making.

Scott [20] argues that there is usually ambiguity about which code, standard, guidance or legislation should be used when it comes to investigation of fire in buildings and decision-making. Globally, many fire safety requirements often conflict with government, professionals and other sectors' interests. For example, the UK Ministry of Housing, Communities and Local Government [21] gave an order after Grenfell Tower tragedy *"that deadly ACM cladding (with uncontrollable fire spread) be removed from all high-rise buildings across the four nationals"*. Three years after the directive, the National Audit Report [22] claimed that more than 300 tower blocks in the UK are still waiting to be stripped of such cladding because of different legislation, local government fire safety guidance, professional recommendations and other sectors' interests.

Moreover, there is a huge dichotomy between fire safety legislation and realistic fire safety decision-making. Contemporary analysis of fire safety laws show that most legislations are often silent in terms of emergency decision-making, specifically during fire scenarios. Ronchi and Nilsson [9] opine that different pieces of fire safety legislation across the globe have common features that include fire protection, detection, warning, emergency and firefighting measures; with little or no emphasis on effective decision-making when fires occur in high-rise buildings

*1.4. Emergency Decision Making (EDM)*

In recent times, emergency decision making (EDM) has become an important research topic because of its significance in remedying adverse safety events such as fire in high-rise buildings. A review of the literature shows that three in five high-rise building fire incidents result in serious fatalities [23]. Westgarth [1] claims that the catastrophic consequences of fire in high-rise buildings suggest that there are significant shortcomings with most EDM. Timely and effective EDM is an important process in emergency management because it helps mitigate risk to loss of lives and properties. However, EDM is often hampered by a lack of information, a lack of knowledge and time pressures with resultant potential consequences Levy [13]. Dong [24] argue that different factors play important role in EDM. For example, good knowledge and understanding of emergency events, psychological behaviour, the magnitude of an event, quick deployment of rescue facilities and access to emergency sites, all contribute significantly to informed firefighting decision-making.

Jonassen [25] posits that decision-making is *"about identifying the benefits and limitations of available courses of action as well as weighing, selecting, and justifying these alternatives"*. Eismann [26] opine that the set goals of emergency services and perceived EDM by decision-makers are always at variance because of complexity that characterises emergency situations. Apart from dealing with complexity, such decision-making can also vary in terms of respective structuredness (i.e., the extent to which problem elements and potential solutions are knowable and understandable to decision-makers) and abstractness (i.e., the extent to which problems are specific to a given context or situation [26,27]).

On the other hand, there is a need to bear in mind that decisions made under conditions of stress and panic usually affect decision outcomes. Comfort [28] and Sellnow [29] argue that decisions made under conditions of stress or tension are typical to crises and emer-

gencies; and such circumstances can challenge decision-makers' sense-making capacities. Decision-making outcomes are deemed to be primarily dependent on the cognitive capabilities of decision-makers. Indeed, EDM is significantly influenced by a decision-maker's experience and knowledge in the decision-making domain. Kapucu and Garayev [30] highlighted four incidental factors that interact with EDM namely:

(i) Individual actors involved in the decision-making;
(ii) Organisational system of emergency management;
(iii) Operational capabilities of the decision-makers;
(iv) Emergency decision environment.

Apart from these highlighted factors, there is a need to consider the "natural and causal" factors of emergency incidents such as the fundamental and causative influence of fire in buildings. Xin and Huang [4] in their publication entitled "Fire risk analysis of residential buildings based on scenario clusters" argue that "though there are array of studies about fire safety in buildings, there is a need to advance research on the proliferation of building materials and their resultant effects on rapid rate of fire spread in high-rise buildings". Arguably, traditional building materials (such as brick, block, and concrete) are more fire retardant compared to many contemporary building materials that strongly support combustion. Scott [31] avows that a thorough understanding of the rate of fire spread in materials is essential for effective emergency decision-making. Please refer to Table S1 in Supplementary Materials.

## 2. Materials and Methods

The study adopts phenomenology as its theoretical basis, underpinned by inductive reasoning; in addition to analysis of three case studies of fire events in high-rise buildings. A research strategy rooted in phenomenology was used because it allows study participants to express their perceptions and expectations based on their individual experiences. Leedy and Ormrod [32] assert that a phenomenological study brings forward the experiences, understanding and perceptions of individuals (about a phenomenon) from their own perspectives. The study data were collected via Focus Group Discussions (FGDs). A total of three FGDs (involving Fire and Rescue Services and occupants of high-rise building with first-hand experience of fire incidents in their buildings/apartments in England and UAE) were considered for interviews and analysis. The choice of England and UAE was informed by recent fire incidents in high-rise buildings in both countries in the last five years.

A purposive (also known as judgement) sampling strategy was used to select eight Fire and Rescue Services professionals (six from England and two from UAE) and six survivors of fire in buildings (five from England and one from UEA) with first-hand experience of fire experiences. The researchers tried to balance the selection of the members in Focus Groups (to include participants that stay-put and those that escaped by running out of the building) from both countries. However, some participants invited for interviews could not participate for various reasons. For balance reporting of events the authors obtained documentary data for robust and true accounts of fire events in buildings.

All fire and rescue operatives interviewed had at least six years or more work experience. Also, high-rise building occupants with first-hand experience of fire incidents were selected based on record of fire and rescue services register in affected areas. These criteria were used to pre-qualify participants for discussion in the FGD. Leedy and Ormrod [32] opine that studying multiple perspectives of a phenomenon can help to generalise the findings of phenomenological studies. However, there are different views regarding the number of participants required in FGD. For example, Babbie [33] states that 12 to 15 people could be brought together to engage in a guided discussion of an issue in a typical FGD. Krueger [34] is of the opinion that 5 to 10 participants should be acceptable for an FGD. For wider participation and validity, a minimum of five and a maximum of 10 participants were considered for each of the FGDs in this study. On average the ratio of fire and rescue services operatives to survivors of fire incidents in high-rise buildings with

first-hand experience of the subject matter was 5:2. All focus group meetings were held using MS Teams between February 2019 to December 2020. All participants were initially contacted via telephone and e-mails; their consent was sought, and an MS Teams meeting was subsequently arranged.

The FGDs discussions and iterations were tape-recorded using audio tape and MS Teams recording devices to allow for easy transcribing. The data obtained were coded and transcribed using both MS Word 2020 document and NVivo 12. The data were analysed using content analysis. The choice of content analysis was informed by the following factors: opportunity to generate useful data that can be used for practical application, the research prospect to develop specific insights, potential to eliminate bias within the data, smaller sample size, and above all the focus group discussions and interactions were open-ended. Research ethics approval was sought from Coventry University Research Ethics Committee and it was granted (see attached ethics certificate in the Supplementary Materials). Specific attention was paid to the data collection techniques, precautionary procedure and data analysis in order to uphold trustworthiness as illuminated in the succeeding sections below.

### 2.1. Data Collection Techniques and Scrutiny of Three Case Studies

First, documentary data relating to emergency services' handling of selected fire incidents in high-rise buildings were collected. Using Freedom of Information requests in the UK, London Fire Bridge and Greater Manchester Fire and Rescue Services were contacted concerning their dairies/logbook entry and real-time radio monologue (communication) regarding the Grenfell Tower fire in 2017 and the University of Bolton student's accommodation fire incidents in 2019. The UAE Fire and Rescue Services also shared real-time radio records regarding the Abbco Tower inferno in Sharjah in the UAE. In addition, two UAE senior fire superintendents with over 11 years' experience provided detailed operational accounts of events that transpired between fire and rescue services' operatives and the Abbco Tower building occupants. Study participants completed research consent forms, but their identities were kept anonymous.

To corroborate some the data, the researchers analysed communication between fire and rescue operatives and building occupants published in the Grenfell Tower Inquiry GTI [19] and radio monologue for the three case studies examined by this study. All case studies (archive) data obtained were carefully scrutinized, with specific emphasis paid to "why" and "how" communications about certain incidents such as record of first distress call to emergency services and approximate time when major incident was declared by emergency services. The researcher cautiously observed key words used in the documentary data or records of survivor's accounts for each case study. For example, key words or phrases such as stay-put, "evacuate", "relocate" "removal strategy", "safe operation", "occupant safety" and timing of control room directives and fire operatives site actions were scrutinised to enable the researchers answer the study research question. Tables 1 and 2 in the following pages present a summary of the three case studies and additional cases of survivor's accounts concerning fire safety experiences in various buildings.

The second phase of the study data collection involved FGDs and iterations. The discussion process was enabled and facilitated with the use of MS Teams, and each FGD lasted approximately 50 to 80 min. All data (question and answer sessions) were recorded via MS Teams and with a digital audio recorder as a back-up device. The interview and discussion data were subsequently transcribed, coded and analysed. The textual contents of the interview data and observed dairies were transcribed into manuscript, input into NVivo 12 software and coded using key study themes. Some of the key themes created to facilitate easy search codes were stay-put, "correctness", "effectiveness", "emergency services", "evacuation", "quality of operation", "conformity to acceptable safety standard", "misjudgement", "magnificent" and "strategy".

**Table 1.** Summary of three case studies of fire in high-rise buildings.

| S/No | Case Studies | Locations/Year of Incident | Nature of High-Rise Building | Height of Building Based on Floors | Cause of Fire | First Call to Emergency Services (Approx. Time) | Major Incident Declared (Approx. Time) | Recorded Communication to Building Occupants on Radio Log and Logbook | Recorded Fatalities/ Injuries | Record of People Safely Evacuated from Building |
|---|---|---|---|---|---|---|---|---|---|---|
| 1 | Grenfell Tower fire | London England—2017 | Residential block of flat | 23 storey | A faulty fridge/freezer in the kitchen | 00:04 a.m. | 02:06 a.m. | 'Stay-put' | 72 Deaths and more than 70 injured | 103 |
| 2 | The Cube: student accommodation at Bolton | Bolton England—2019 | Student accommodation | 7 storeys | Discarded cigarette | 08:29 p.m. | 08:34 p.m. | 'evacuate instantly'; 'find your way out immediately' etc | 27 injuries | 107 |
| 3 | Abbco Tower | Sharjah Dubai, UAE—2020 | Residential block of flat | 49 storey | Discarded cigarette butt or shisha coals | 9:00 p.m. | 9:02 p.m. | 'Out of the building'; 'go out immediately' | 27 injuries | 12 |

**Table 2.** Summary of the accounts of fire survivors' who adopted a stay-put strategy.

| S/No. | Summary of Survivor's Narrative of How They Escaped Adverse Fire Incidents in Buildings | Frequency Survivor's Narrative | Summary of Other Comments Made by Building Fire Survivors |
|---|---|---|---|
| 1 | Occupants and firefighter Communication | 17 | " … the scale of fire was small … " |
| 2 | Timely intervention of Emergency Services | 21 | " … it was scary; I thought I was going to die … " " … timely intervention of emergency services saved us … " |
| 3 | Fire Drill and Escape signs | 9 | " … my greatest problem was smoke inhalation … " |
| 4 | Nature of Fire (Minor fire) | 29 | " … I pray it never happens again … " " … worst experience in my life … " |
| 5 | Apartment (flat) proximity to ground floor | 11 | " … it was a life or death situation … " " … it was confused situation … " |

*2.2. Trustworthiness and Validity of Data*

The researchers were mindful of the endless theoretical arguments about the validity of qualitative inquiry, often defined as "truth", credibility or "integrity of qualitative inquiries" [35,36]. To avoid philosophical arguments about the validity of the study qualitative data the authors accept the standpoint of Kuzmanić [37] assertion that " ... there is a pure 'form of truth' somewhere out there, which can be discovered (through construct, external and internal validity) using appropriate and, most importantly, valid research methods ... " For straightforwardness, the researchers infer valid qualitative research (interview and iterations data) to represent credible social worlds (construct) or different interpretations of words for the benefit of the readers.

Therefore, the validity of the phenomenological inquiry was addressed through three fundamental areas: production (the design of the interview questions, the interview process, probing and interrogation of interviewees and the recording of the data), presentation (replicability, valid inference and arrangement of the data) and interpretation (meaningful discussion of data). The choice of content analysis was explained in Section 3 of this report. For example, the interviewees in the FGDs were presented with fire and rescue services dairies, logbook entry and real-time radio monologue (communication) regarding fire incidents in three high-rise buildings. Subsequently, the interviewees were asked series of interview questions; response to each question was afterward followed by probing and detailed interrogation to ascertain robustness, trustworthiness and reasonableness of answers and interactions of study participants.

Below are some of the probing questions designed for the study, together with some key themes from the interviews and dairies of fire and rescue services. The interviewees' responses (data) were trimmed and summarised for better understanding and spontaneity of the interaction between the study participants and the researchers. Some textual excerpts are presented verbatim as illustrated below for a better understanding of the participants' viewpoint regarding the study aim.

## 3. Results

**Researcher question:**

**Question 1a: As an experienced fire and rescue officer what is your view about the stay-put instruction (a misjudgement or magnificent strategy) when dealing with fire safety in high-rise building?**

**Responses from interviewees**

*" ... my work experience tells me that stay-put tactic is a good strategy ... that is dependent on the nature of fire and building ... it is a suitable approach if the fire is contained within a short period of time say five to 10 min ... "* —**Firefighter II Officer in UAE.**

Similar opinions occurred three times in the discussions.

*" ... I don't have a clear-cut answer to your question, ... because there are instances where Stay-Put instruction led to disaster ... and in other cases it was deemed a perfect strategy ... "* —**Sub Officer I–London.**

Similar opinions occurred twice in the discussions.

**Follow-up and probing questions**:

For clarity and better understanding of the focus group discussions regarding question 1 above; the following questions was asked.

**Question 1b: Can you explain further; what do you mean by saying that the stay-put tactic is a good strategy?**

*" ... stay-pu' is a good fire safety tactic if the fire is not spreading rapidly and uncontrollably ... for example, if there is a fire in second floor of a building that can be contained easily ... occupants in other floors do not need to evacuate ... I will insist that the occupants remain where they are ... to avoid commotion mainly if the building does not have good emergency exit facilities ... "* —**Chief Fire Officer Manchester.**

Similar opinions occurred twice in the discussions.

*" . . . perhaps, misjudgement or magnificent are not the right words to describe effectiveness of stay-put instructions . . . reason being that fire safety outcome or successful evacuation of occupants from a high-rise building is subject to many factors . . . "* — **Station Officer Manchester.**

Similar opinions occurred twice in the discussions.

**Question 1c: Can you identify factors that influence the success of stay-put tactic please?**

*" . . . nature and age of the building . . . for example, our record show that 2 in 5 high-rise buildings in our metropolis have poor fire emergency exits, other factors are occupants in a building when a fire occurs . . . height of the building, . . . emergency services preparedness and access to the building, . . . time of the day when the fire occurs . . . cladding fabric of a building . . . these factors certainly influence successful evacuation or outcome of stay-put tactic . . . "*

**—Fire and Rescue Superintendent Dubai**

Similar opinions occurred five times in the discussions.

*" . . . firefighters and building occupants training and awareness . . . effectiveness of fire safety devices* i.e., *fire alarm, fireproof doors, emergency exist accessibility . . . indeed, building occupants readiness to obey stay-put instruction is another factor that is often neglected . . . "* **—Station Commander.**

Similar opinions occurred three times in the discussions.

Thus far, the questions and discussions above provide some understanding regarding the experience of emergency services mode of operation, effectiveness of the stay-put tactic, factors that are likely to influence success of stay-put guidance, and so on. While there is no reason to doubt discussions and interactions that transpires between fire and rescue services and the researchers; there is a saying that "no man can be a judge in his own case". Therefore, for objectivity and thorough understanding of the study research question; survivors of fire incidents in high-rise buildings with first-hand experience concerning the stay-put tactic were interviewed as part of the FGDs. Below are some textual questions and quotations from some fire survivors for better understanding of their viewpoints:

**Question 2a:** From the research participant invitation and consent form you claimed that you are one of the survivors of fire incident in a high-rise building. **Can you tell this forum your personal experience concerning the emergency services' stay-put instruction when fire occurred in your building?**

*" . . . my experience is indescribable with great distress when the fire occurred . . . I was on the 3rd floor . . . with my floormate screaming Fire, fire!!! . . . out, out !!!, check on neighbours !!!, . . . out, out!!, . . . my recollection was an experience characterised by confusion, fear, commotion . . . "* **—survivor of the University of Bolton Hall of Resident–2019.**

Similar opinions occurred twice in the discussions.

*" . . . my experience does not encourage occupants to listen to emergency services stay-put instruction, . . . it is a state of confusion . . . that coerce occupants to do anything possible to find safety . . . those of us that ran out quickly regardless of firefighter's instructions and fellow occupant yelling, and cry survived . . . my next-door neighbor died . . . I cannot tell whether she obeyed stay-put directive or not . . . ."* **—survivor of Grenfell Tower fire 2017.**

Similar opinions occurred twice in the discussions.

**Follow-up probing questions:**

**Question 2b: Tell us your personal fire experience in high-rise building; do you agree with fire and rescue officers' claim in this discussion that stay-put is a good fire safety strategy?**

" . . . *certainly not . . . it is a misjudgement . . . with little regard for psychology of building occupants' and unpreparedness of true fire situation . . .* " —**survivor of Grenfell Tower fire 2017.**

Similar opinions occurred three times in the discussions.

" . . . *stay put is not a strategy, the tactic has no rational plan with a clear outcome . . . it is difficult to determine a clear outcome from stay-put instruction . . . besides most occupant hardly obey such order . . . stay-put instruction has no clear method or logical tactics that guarantee positive outcome . . . there is no evidence that occupants who obeys such instruction will be rescued successfully . . . therefore, in strict terms it is not a strategy . . .* " —**survivor of Abbco Tower fire 2020.**

Similar opinions occurred three times in the discussions.

**Question 2c: It may be wrong to describe the stay-put order as a misjudgement. Why do you think that it is a misjudgement?**

" . . . *fire and rescue services' claims in these discussions clearly suggest that success of stay-put is dependent on many factors such as nature of fire, building, rate of fire spread, escape routes, etc . . . these factors are not directly controlled by fire and rescue operatives . . . hence, how do you plan for something that is not directly under your control . . . strategy is all about planning, review and control . . . I am not convinced that stay-put tactic is a magnificent strategy . . . because it is not underpinned by a clear plan and definitive outcome . . .* " —**survivor of the University of Bolton Hall of Resident–2019.**

Similar opinion occurred twice in the discussion.

" . . . *available evidence suggests that stay-put tactic is only useful in small scale fire . . . medium and large-scale fires in high-rise building must be followed by instruction for occupants to evacuate immediately . . . the sole purpose of fire safety in building is to protect lives and property and not to make bad situation worse* via *vague instruction . . .* " —**Leading Firefighter Manchester.**

Similar opinions occurred twice in the discussions.

Apart from the above questions and some study participants were also interrogated on other issues relating to construction materials, fire regulations and design of high-rise buildings which constitute rich source for other papers. However, the concluding part of the FGDs and interviews focused on how to achieve effective and sustainable communication from the fire and rescue services that engenders safety and the smooth evacuation of occupants from high-rise buildings. Thus, participants were questioned and cross-examined as follows:

**Question 3a: What message should emergency services convey to enhance safety and the smooth evacuation of occupants during fire in high-rise buildings?**

" . . . *the downside of stay-put tactic is that it is imprecise in terms of achieving a desirable outcome . . . it can also lead to confusion to both firefighters and occupants . . . the correct message to occupants during emergency is 'find your way out immediately'; . . . 'evacuate instantly' . . . particularly in buildings with poorly designed egress and inflammable construction materials . . .* " —**Fire and Rescue Officer in London.**

Similar opinions occurred three times in the discussions.

" . . . *stay-put strategy is not completely wrong per say . . . if the fire is containable within a flat and if a building has good fire-retardant fabric . . . outside these parameters the correct message to occupants must be clear and precise . . . 'abandon whatever you are doing; and leave the building using the nearest exit' . . .* " —**Fire Station Officer in Manchester.**

Similar opinions occurred four times in the discussions.

" . . . *stay-put order is certainly not a good fire safety tactics . . . because fire in buildings and its outcome cannot be predicted . . . therefore the correct message to occupants must*

*be unpretentious and clear . . . 'abandon whatever you are doing leave the building immediately using available exit' . . . "* **—survivor of Grenfell Tower fire 2017.**

Similar opinions occurred twice in the discussions.

Summary of discussions and suggestions thus far concerning how to achieve effective and sustainable fire safety communication between emergency services and occupants seem to focus on "evacuate instantly" and "abandon whatever you are doing; and leave the building". Probing question: **What is your advice for high-rise building occupants who are vulnerable, i.e., children, elderly or sick people who obey the stay-put instruction?**

*" . . . existing fire safety communication techniques i.e., fire alarm systems are old-fashioned . . . fire alarm systems need to be adapted for real-time targeted communication . . . we also need artificial intelligence communication system that entails computerisation of building occupant's movement using image detector to enhance fire and rescue services communication strategy and effective evacuation of vulnerable folks during fire . . . "* **—Firefighter London.**

Similar opinions occurred six times in the discussions.

*" . . . 'stay-pu' strategy should not only be use for occupants that are vulnerable . . . that is for controllable fire in a building . . . best advice is to have an up to date register and viable apartment allocation policy . . . for example, allocation of ground and first floors in a high-rise building to vulnerable people . . . to allow easy and faster rescue operation . . . "* **—Chief Fire Officer.**

Similar opinions occurred five times in the discussions.

*" . . . I support the use of . . . up-to-date identification record, prioritisation of vulnerable people in floor allocation, use of computerised image detectors . . . but we have to consider data protection laws . . . "* **—Fire and Rescue Officer London.**

Similar opinions occurred four times in the discussions.

Tables 1 and 2 provides summary of building fire survivors' narratives that adopted the stay-put tactic. A total of 11 building fire cases studies (seven in England and four in UAE) were scrutinised for thorough understanding and balance viewpoints of survivors that adopted stay put and non-stay put tactics. Note that the findings in Table 2 represent narratives of survivors that adopted the stay-put tactic only. The data covers various fire incidents in high-rise building spanning over 20 years. Leedy and Ormrod [32] stress that the choice of phenomenological research is usually informed by its potentials to help describe and explain essence of a phenomenon by exploring perspective of those who have experienced a particularly problem or by examining case studies and archive data.

Therefore, the authors relied on Fire and Rescue Services dairies to ascertain true accounts of building fire survivors that adopted stay put tactic. The decision to use both archive (true account/narratives of survivors that adopted the stay-put tactic via Fire and Rescue Services diaries); and case studies data that mirrors survivors that adopted the stay-put tactic was due to failure on the part of survivors that adopted stay put to attend organised FGDs and interviews. Thus, for balance reporting and to negate bias regarding the study findings the authors relied on archive and documentary data about survivors that adopted the stay-put tactic.

Moreover, the credibility of findings in Table 2, that is, the viewpoint of survivors that adopted the stay-put tactic, is enhanced by repeatability, frequency of survivors narratives, large population sample of data analysed and consistency of survivor's story spanning over a 20 year period; compared to non-stay-put accounts based on interview and FGDs. Besides, detail review of archive and documentary data suggest that elderly, vulnerable people and high-rise building occupants that lives in uppermost floors are likely to adopt stay-put tactics.

## 4. Discussion and Findings

Fundamental lessons from the study literature suggest that the stay-put strategy used by fire and rescue services is not thoroughly understood. Most literature suggest that the

stay-put tactic is a magnificent practice if coordinated properly. Perhaps, the stay-put tactic has potentials to protect, control and facilitate orderly evacuation of occupants in minor fire incidents; particularly in high-rise buildings. Conversely, some study participants believe that the stay-put concept is an obsolete guidance that should not be used in fire evacuation operation because of its imprecise and unclear outcomes. The later argument appears to be watertight, because the history of fire in modern high-rise buildings paints a picture of rapid fire spread due to multiple factors relating to modern construction materials, poor egress design, and inability to predict outcome of the stay-put tactic. It is important to stress that, though, the sources of data about survivors that adopted the stay-put tactic (i.e., data obtained via log-book and Firefighter dairies) and those who did not (i.e., data obtained via interview/FGDs) are different; the veracity of both sources are credible and adequate to deduce robust conclusion about the research problem. Majority of study participants believe that the stay-put instruction is inadequate; when a full-blown fire occurs in modern high-rise buildings. The findings from FGDs are consistent with three case studies scrutinised as illustrated in Table 1 above. A common deduction from the three case studies reveal that the fire spread rapidly and uncontrollably in the buildings examined. However, timely communication and swift emergency decision making (swift declaration of major incidents) in both Abbco Tower and the Cube building adverted fatalities; though minor injuries were recorded compared to deadly outcome in the Grenfell Tower incident.

Furthermore, three in every five study participants in the FGDs and interviews agree that 'Stay-Put' tactic is likely to result in disastrous outcome; because of confusion and panic resulting from such instruction. In terms of the study research question, there were polarised views among study participants that 'stay-put' is not a strategy; due to lack of rational plan and immeasurable safety outcome. For example, all study participants agree that in most cases building occupants hardly obey 'stay-put' directive. Besides, the effectiveness of 'stay-put' instruction as an emergency fire safety plan is dependent on factors such as rapid rate of fire spread, nature of fire, access to emergency escape routes, etc. Arguably, these factors are rarely under the control of fire and rescue services. Thus, it is illogical to describe 'stay-put' as a magnificent strategy.

However, there was unanimous consent among study participants regarding how to achieve effective and sustainable fire safety communication that engender safe and smooth evacuation of occupants from high-rise buildings. The study participants identified factors such as compulsory and regular fire safety communication training, computerisation of building occupant's movement using image detector, and allocation of lower and accessible floors to vulnerable people as future requirements to minimise fire fatality in high-rise buildings.

## 5. Conclusions

Emergency decision-making is an important research topic in recent years. However, there is little awareness about fire and rescue services' emergency decision-making; particularly regarding effectiveness of the stay-put tactic. Review of literature suggest that this study is the first of its kind to examine impact of fire and rescue services communication and emergency decision-making, using non-numeric data to understand the concept of 'stay-put' tactic. The validity of the study findings stems from first-hand accounts of both high-rise building fire victims and seasoned emergency services narratives. Also, the strength of the study can be drawn from the systematic collection, organisation, description, and interpretation of rich textual data that contributes to thorough understanding of 'stay-put' phenomenon.

Generically, there is overwhelming belief by professionals that certain fire safety guidance and practices such as the stay-put tactic is obsolete; with potentials to cause catastrophic misjudgement especially during conflagration in high-rise buildings. Moreover, recent GTI [2] report clearly berated the stay-put tactic. Yet, findings from this study reveal that many fire and rescue services operatives interviewed, concur that 'stay-put' guidance

is an integral part of their training and modus operandi. Archive evidence show that 'stay-put' tactic is institutionalised in most fire and rescue services training and operations. Arguably, continuous use of the stay-put strategy has potential to engender early and safe evacuation of occupants from high-rise buildings. For example, three out of five fire and rescue officers interviewed directly acknowledged that 'stay-put' tactic is mostly likely to constitute a misjudgement. Some firefighters claim that, and I quote " . . . 'stay-put' principle is undoubtedly unsuccessful in an overwhelming number of fire incidents in modern blocks of flats . . . ". Furthermore, cross-examination and probing of seasoned fire and rescue operatives reveal that " . . . in first instance, the role of firefighters is to safely evacuate everybody from a building . . . 'stay-put' tactic should be used only when a fire incident is deemed controllable and containable . . . ". Indeed, this quote encapsulate the research question on whether 'stay-put' is a misjudgement or magnificent strategy. However, the unanswered question is when is a fire deemed controllable and containable considering the scale of fuel loading in different construction materials? Moreover, six out of eight Fire and Rescue Services professionals interviewed concurred that the use of artificial intelligence communication systems and infrared image detector cameras are required to enhance quicker and smoother fire safety evacuation in high-rise buildings. They also confirmed that development of artificial intelligence and infrared image detectors cameras with potentials to enhance communications and movement of occupants in high-rise buildings are currently in pilot or batch stage. Finally, there is need for future research to measure effectiveness of artificial intelligence, use of infrared image detector cameras and privacy of building occupants.

## 6. Patents

There are no patents resulting from the work reported in this manuscript.

**Supplementary Materials:** The following are available online at https://www.mdpi.com/article/10.3390/buildings11080339/s1.

**Author Contributions:** The following are categorisation of individual contribution to this paper: Conceptualization of rsearch idea: A.O.A. Research methodology: A.O.A., A.A., C.N.; Use of software: A.O.A., A.A., D.J.E.; Validation of research data and findings: A.O.A., A.A., C.N. Formal data analysis: A.O.A., A.A. Investigation: A.O.A., A.A. Research resources: A.O.A., A.A. Research data curation: A.O.A., A.A., C.N.; Writing—original draft preparation: A.O.A. Writing—review and editing: A.O.A., D.J.E. Visualization: A.O.A.; Supervision: A.O.A., D.J.E.; Project administration: A.O.A.; D.J.E.; Funding acquisition: A.O.A. All authors have read and agreed to the published version of the manuscript.

**Funding:** This research did not receive external funding. The research was a team work from University of East London, Coventry University and Birmingham City University.

**Institutional Review Board Statement:** The study was conducted according to the guidelines of the Declaration of Coventry University and was approved by Coventry Research Ethics Committee.

**Informed Consent Statement:** Informed consent was obtained from all subjects involved in the study.

**Data Availability Statement:** Data regarding interviews and various Focus Group Discussion can be found via: https://web.microsoftstream.com/studio/meetings (accessed on 14 September 2020).

**Acknowledgments:** The authors want to acknowledge organisations and high-rise building survivors that participated in the study.

**Conflicts of Interest:** The authors declare that there is no conflict of interest in this report.

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
