# Peer review of "Fire Safety in High-Rise Buildings: Is the Stay-Put Tactic a Misjudgement or Magnificent Strategy?"

_buildings, doi:10.3390/buildings11080339_

Round 1

Reviewer 1 Report

The major problem of this research is that the authors did not interview the survivors who adopted stay-put strategy during fire. Since all interviewed survivors escaped out of the building during fire, they of course would not agree on stay-put. Therefore, the conclusion has bias unless someone adopted stay-put also discommend this strategy.

Secondly, the abstract highlighted the artificial intelligent communication systems. However, only one fire fighter proposed this system in his response. This conclusion is also not well justified.

Thirdly, only survivors from three fire accidents were interviewed. But the total no. of interviewees was not reported. The survey with limited no. of survivors did not reflect the general views of the majority.

More data collection is required to cover both adopted stay-put and run-away survivors to arrive a more neural conclusion.

Author Response

Reviewer 1 summary of report form:

English language and style

(x) English language and style are fine/minor spell check required

  1. Does the introduction provide sufficient background and include all relevant references? -YES
  2. Is the research design appropriate? - - Must be improved
  3. Are the methods adequately described? - Can be improved
  4. Are the results clearly presented? - Must be improved
  5. Are the conclusions supported by the results? - Must be improved

Reviewer 1: comments and Suggestions for Authors

  1. The major problem of this research is that the authors did not interview the survivors who adopted stay-put strategy during fire.

Response from authors:

Section 2.0 with subheading “Materials and Methods” and continuous line 286 and 296 the authors made it clear that the study adopted phenomenology type of research, underpinned by inductive reasoning; in addition to analysis of three case studies of fire events in high-rise building”

First, survivors who adopted stay-put strategy during fire were invited severally for interview and focus group discussions. But these categories of interviewees were unable to attend series of focus group discussion and interviews.

However, the authors relied on building fire survivor’s accounts extracted from logbooks for balance reporting and analysis of “stay-put” tactics. See continuous lines 597 to 612 for detail reporting of building fire survivor’s accounts.

Besides, the initial intention of the authors was to develop a separate paper based on Building fire survivor’s accounts in the Journal of Buildings. However, for balance reporting a brief account of survivors who adopted stay-put strategy during fire have been reported in this paper. See continuous line 597 to 612. The findings are based purely on three case studies.

  1. Secondly, the abstract highlighted the artificial intelligent communication systems. However, only one fire fighter proposed this system in his response. This conclusion is also not well justified.

Response from authors:

In section 2.0 Materials and Methods between continuous line 296 to 299 the authors made it clear that A purposive (also known as judgement) sampling strategy was used to select eight Fire and Rescue Services professionals (six from England and two from UAE) and six occupants (Five from England and one from UEA) of high-rise buildings with first-hand experience of fire experiences”.

The six out of eight Fire and Rescue Services professionals interview concurred that the use of artificial intelligent communication systems and infrared image detectors camera are required to enhance quicker and smoother fire safety evacuation in high-rise buildings.

Senior fire and rescue operatives interviewed also confirmed that development of artificial intelligent and infrared image detectors camera with potentials to enhance communications and movement of all occupants in high-rise buildings are currently in pilot or batch stage.

Please refer to continuous lines 567 to 591 and 680 to 688 for detail reporting of findings and conclusion sections regarding the use of artificial intelligent communication systems and infrared image detectors camera.

  1. Thirdly, only survivors from three fire accidents were interviewed. But the total no. of interviewees was not reported. The survey with limited no. of survivors did not reflect the general views of the majority.

Response from authors:

The research paper adopted a phenomenology research method. The research method often relied upon description of accurate person’s ‘lived’ experience in relation to what is being studied.

In this study, though survivors that adopted “stay-put” were not interviewed their personal account was clearly captured and recorded based on Fire and Rescue Services documentary data.

See continuous line 594 to 611 for balance reporting and brief account of survivors who adopted “stay-put” strategy during fire in buildings.

  1. More data collection is required to cover both adopted stay-put and run-away survivors to arrive a more neural conclusion.

Response from authors:

The study provided robust data that covers participants that adopted ‘stay-put’ and run-away survivors. Findings regarding survivors that adopted ‘stay-put’ strategy are presented in continuous line 597 to 612.

Finally, the entire work has been proofread to avoid grammar errors.

Reviewer 2 Report

This is a good manuscript working on the major issue. The authors clearly summarized their work and results. However, some link to other disasters (other than fire, such as earthquakes, floods, landslides, volcanoes etc.) preparedness and pieces of evidence on citizen's knowledge to respond to the event would better to include.

In any disaster event, risk communication plays a major role in the response by citizens. It solely depends on their perception and knowledge to anticipate the risk in a short span of time. Authors can find an example of an earthquake event where a person lost her life due to a misunderstanding of risk and more (https://doi.org/10.1186/s40677-020-00150-2). 

During any fire incident, residents (or other users) should know about the potential risk, evacuation route, and a proper decision on the basis of their knowledge. They should be aware of the fire alarm system, fire extinguishers, hydrants and other measures placed in the building. First responders are always citizens in the building; evacuation methods, 'stay-put' or any other should be based on the situation. Hence, risk communication and awareness among the users should also be focused on.

Authors are advised to improve their literature review part connecting the similar scenario of other disaster events where the 'standard evacuation system' is not properly working.

Author Response

Reviewer 2: summary of report form:

Reviewer 1: Comments

English language and style

(x) English language and style are fine/minor spell check required

  1. Does the introduction provide sufficient background and include all relevant references? – can be improved
  2. Is the research design appropriate? - - YES
  3. Are the methods adequately described? - YES
  4. Are the results clearly presented? - can be improved
  5. Are the conclusions supported by the results? - can be improved

Finally, the entire work has been proofread to avoid grammar errors.

Reviewer 2: comments and Suggestions for Authors

  1. This is a good manuscript working on the major issue.

Response from authors:

The comment above is affirmative (positive) – Thanks

  1. The authors clearly summarized their work and results. However, some link to other disasters (other than fire, such as earthquakes, floods, landslides, volcanoes etc.) preparedness and pieces of evidence on citizen's knowledge to respond to the event would better to include.

Response from authors:

First thanks for your suggestion for authors to link the paper to other natural disasters such as earthquakes, floods, landslides, etc. However, after careful review of the comment; the authors decided not to link the paper to natural disasters.

The reasons are: fire in buildings are often man-made disasters or often caused by errors. Second, preparedness strategy and disasters environment for man-made and natural disaster are usually different. Third, most natural disasters such as earthquakes, floods, landslides, etc hardly result in fire and resultant adoption of “stay-put”.

Therefore, linking fire in buildings to natural disasters will complicate the entire paper. However, the authors alluded to terrorism attack such as World Trade Centre Twin Towers in New York City that resulted in huge fire. See continuous line 41 to 45.

  1. During any fire incident, residents (or other users) should know about the potential risk, evacuation route, and a proper decision on the basis of their knowledge. They should be aware of the fire alarm system, fire extinguishers, hydrants and other measures placed in the building. First responders are always citizens in the building; evacuation methods, 'stay-put' or any other should be based on the situation. Hence, risk communication and awareness among the users should also be focused on.

Response from authors:

The paper clearly identified potential risk, evacuation route, and proper decision that building occupant should follow when fire breaks out in a building. See continuous line 130 to 147. Moreover, the word “communication” was used 18 times in the paper and it clearly focused on risk communication and awareness. The paper clearly put specific emphasis on risk communication and awareness. See continuous line 61 - 64, 90 - 93, 130 - 146, 571 – 580,  610 – 621 and 680 – 687.

  1. Authors are advised to improve their literature review part connecting the similar scenario of other disaster events where the 'standard evacuation system' is not properly working.

Response from authors:

Please refer to Reponses to Reviewer 2, comment 1 above. Reasons for not linking the paper to natural disaster are highlighted above. The authors strongly believe that linking the literature review to nature disaster will be odd. 

Finally, the entire work has been proofread to avoid grammar errors.

Reviewer 3 Report

This manuscript discussed an interesting topic"stay-put" using phenomenological research strategies. Some comments are as follows.

  1. Where is Figure 1 from? Reference 17? This reference focused on natural disasters. I could not see the description to discuss the six fundamental pillars of fire safety in [17].  Additionally, the six pillars may not be recognized by most fire researchers.
  2. I think fire compartmentation is an important issue when making evacuation decisions. If the fire compartmentation design is good, it may be ok to stay-put.
  3. This study interviewed 8 fire rescues services professionals and 6 occupants from England and UAE. First, 8 and 6 people may not be adequate. Second, the fire codes related to fire evacuation need to be compared, otherwise, the rescue professionals and occupants may have different concepts when making decisions.    
  4. The authors of [4] should be Xin and Huang.
  5. Too may grammatical errors. For example. a "s" should be put after a verb when the subject is third person singular. Some other mistakes exist such as "et al.", "i.e.". The authors need to check carefully. 

Author Response

Reviewer 3: comments and Suggestions for Authors

  1. Where is Figure 1 from? Reference 17? This reference focused on natural disasters. I could not see the description to discuss the six fundamental pillars of fire safety in [17].  Additionally, the six pillars may not be recognized by most fire researchers.

Response from authors:

The reference was wrong. The correct reference for [17] is from the International Fire Protection Journal. Please refer to body of references to see the correct [17] which is

[17] Antell, J. H. (2019) Fundamentals of high-rise fire safety. Journal of International Fire Protection (JIFP). April 2019. Available in https://ifpmag.mdmpublishing.com/fundamentals-of-high-rise-fire-safety/. See continuous line 768 – 770.

Moreover, the six pillars are recognised internationally: See the following publications and journal:

1.      Title of publication: “Fire Safety Design and Evacuation Strategies for High Rise Buildings” Available in https://mfpa.com.my/2021/03/17/fire-safety-design-and-evacuation-strategies-for-high-rise-buildings/

2.      Title of publication “Fire Service Features of Buildings and Fire Protection Systems” published by Occupational Safety and Health Administration (OSHA) available in https://www.osha.gov/sites/default/files/publications/OSHA3256.pdf

3.      Title of publication: “the International Code - 2018 International Fire Code” published by INTERNATIONAL CODE COUNCIL, INC. Available in ISBN: 978-1-60983-739-6.

  1. I think fire compartmentation is an important issue when making evacuation decisions. If the fire compartmentation design is good, it may be ok to stay-put.

Response from authors:

The author totally agrees with reviewers’ comments above and made similar allusion in the discussion and conclusion sections in the paper. See continuous line 57 – 59, 198 -200; 131 – 134; and 195 – 196;

  1. This study interviewed 8 fire rescues services professionals and 6 occupants from England and UAE. First, 8 and 6 people may not be adequate. Second, the fire codes related to fire evacuation need to be compared, otherwise, the rescue professionals and occupants may have different concepts when making decisions.    

Response from authors:

First, the nature of the research problem (fire in buildings is not a daily occurrence) and research methods adopted (phenomenology research and analysis of three case studies of fire events in high-rise buildings) significantly limits the total number of participants interviewed. Besides, the strength of the research method adopted is hinged on the use of phenomenology research and analysis of three case studies.

Moreover, fire codes relating to fire evacuation have been reviewed and compared. See continuous line 153 to 156.

  1.  The authors of [4] should be Xin and Huang.

Response from authors

Reference Xin and Huang (2013) has been checked and reported correctly. See continuous line 46, 53, 109, 276 and 728.

  1. Too many grammatical errors. For example. a "s" should be put after a verb when the subject is third person singular. Some other mistakes exist such as "et al.", "i.e.". The authors need to check carefully.

Response from authors

The entire work has been proofread to avoid grammar errors. Also, the word or phrase "et al." was removed from the entire paper.

Finally, the entire work has been proofread to avoid grammar errors.

Round 2

Reviewer 1 Report

Based on the revisions and responses, please find below the comments:

  1. Our major concern is that the authors did not interview the survivors adopted ‘stay-put’ strategy.
  2. The authors responded “… survivors who adopted stay-put strategy during fire were invited severally for interview and focus group discussions. But these categories of interviewees were unable to attend series of focus group discussion and interviews.”
  3. That means the authors confirmed they did not interview the survivors adopted ‘stay-put’. The authors relied on the log-book information from fire and rescue service.

The data collection approaches for survivors of ‘stay-put’ (by log-book record) and ‘non-stay-put’ (by interview) are different. It is not convincing to arrive a fair judgement. One possible way, although not ideal, is to send the summary shown from line 597 to  599 to the survivors of ‘stay-put’ to see if they have any objection to it.

Reviewer 3 Report

I am fine with the modification the authors have made. 

Author Response

Reviewer 3 is fine or happy with the modification that the authors have made.